# A Systematic Review on Clinical and Health-Related Quality of Life Outcomes following Total Gastrectomy in Patients with Hereditary Diffuse Gastric Cancer

**DOI:** 10.3390/cancers16030473

**Published:** 2024-01-23

**Authors:** Hui Jun Lim, Massimiliano di Pietro, J. Robert O’Neill

**Affiliations:** 1Early Cancer Institute, University of Cambridge, Cambridge CB2 0XZ, UK or huijun.lim@mohh.com.sg (H.J.L.); md460@cam.ac.uk (M.d.P.); 2Department of Sarcoma, Peritoneal and Rare Tumours (SPRinT), Division of Surgery and Surgical Oncology, National Cancer Centre Singapore, Singapore 168583, Singapore; 3Cambridge Oesophagogastric Centre, Cambridge University Hospitals NHS Foundation Trust, Cambridge CB2 0QQ, UK; 4Institute of Genetics and Cancer, University of Edinburgh, Edinburgh EH4 2XR, UK

**Keywords:** hereditary diffuse gastric cancer, prophylactic gastrectomy, surgical outcomes, complications, health-related quality of life

## Abstract

**Simple Summary:**

Hereditary diffuse gastric cancer (HDGC) is associated with early onset diffuse gastric cancer. Definitive treatment is surgery to remove the stomach, which has potential long-term effects on health and quality of life. As such, regular endoscopies may be offered as an alternative. This systematic review aims to evaluate outcomes following surgery for HDGC patients that will aid in patient management. Three hundred and fifty-three patients were examined in 15 studies that reported surgical outcomes. The major complication and mortality rates were 19.2% and 0.3%, respectively. Common complications included wound infection (4.8%), anastomotic leak (4.5%) and lung complications (4.5%). Following surgery, 88.6% of patients had early lesions amongst 414 patients with no lymph node involvement. There was a wide range of psychosocial effects following surgery closely related to the physical symptoms. Overall, surgery is a safe option, and it is important to be under the care of a multidisciplinary team.

**Abstract:**

Hereditary diffuse gastric cancer (HDGC) is an autosomal-dominant syndrome associated with early onset diffuse gastric cancer. Definitive treatment is prophylactic total gastrectomy (PTG) associated with significant morbidity. Studies published from January 2000 to December 2022 reporting clinical, histopathological or health-related quality of life outcomes in HDGC patients undergoing PTG were identified. The study quality was assessed by the “Newcastle–Ottawa scale”. Of the 257 articles screened, 21 were selected. A total of 353 patients were examined in 15 studies that reported surgical outcomes. The median age was 42 years old. The median major complication and mortality rates were 19.2% and 0.3%, respectively. The most common complications were wound infection at 4.8% followed by anastomotic leak and pulmonary complications at 4.5% each. Following PTG, 88.6% of patients had early lesions amongst 414 patients. The mean/median number of signet ring cell carcinoma foci in the gastrectomy specimens was from 2 to 78. All cases were stage 1 with no lymph node involvement. There was a wide range of psychosocial effects following PTG closely related to the physical symptoms. It is imperative for patients to receive comprehensive preoperative counselling to make an informed decision and be followed up under the care of a multidisciplinary team.

## 1. Introduction

Hereditary diffuse gastric cancer (HDGC) syndrome was first described in three extended New Zealand Māori families in 1998 and is characterised by a high incidence of diffuse-type gastric adenocarcinoma at 67% and 83% for males and females, respectively, by 80 years old and an elevated risk of lobular breast cancer in females of up to 55% [1,2]. Up to 3% of all patients who develop gastric cancer have HDGC [3]. This autosomal-dominant condition is predominantly attributed to germline loss-of-function mutations within the tumour suppressor gene *CDH1* [4]. In addition, mutations in a second adherens junction protein, α-catenin (*CTNNA1*), have been reported in a minority of HDGC cases with an uncertain risk in the development of DGC [4]. Both genes play a role in epithelial cell adhesion [5,6]. The mean age of onset of diffuse gastric cancer is 38 years old in patients with HDGC but can vary widely, with cases reported between 16 and 82 years old [7]. The International Gastric Cancer Linkage Consortium (IGCLC) has provided guidelines that include indications for germline genetic testing and counselling in individuals who meet specific criteria based on family history and age at diagnosis [8].

Given the poor prognosis of symptomatic diffuse-type gastric cancer and the high cumulative lifetime risk of developing this, patients with HDGC are commonly counselled to undergo prophylactic total gastrectomy (PTG) at diagnosis and ideally between the ages of 20 and 30 years when the operative risk is likely to be lowest [7]. However, total gastrectomy is associated with considerable morbidity and potential long-term nutritional and metabolic effects that have an adverse impact on the quality of life [9,10]. The evaluation of preoperative nutrition and education regarding post-gastrectomy dietary changes is essential, and all patients proceeding to PTG should have a baseline endoscopy before surgery to determine if there is an established malignancy that would require staging and may warrant neoadjuvant treatment prior to radical total gastrectomy [7]. Although there have been several small single-centre cohort studies examining the outcomes following PTG, a wide range of outcomes have been described. In this systematic review, the clinical and health-related quality of life (QOL) outcomes following prophylactic gastrectomy for HDGC patients are summarised. This will help to guide clinicians in the management of HDGC patients and aid in informed consent-taking.

## 2. Materials and Methods

### 2.1. Literature Search and Study Selection

All articles reporting surgical, histopathological or health-related QOL outcomes in patients with HDGC who underwent total gastrectomy for prophylactic intent only (PTG) were identified via a systematic search of the PubMed, Medline and Scopus databases and the Cochrane Library. Articles published between January 2000 and December 2022 were selected using search terms including “*CDH1* pathogenic variants”, “hereditary diffuse gastric cancer”, “prophylactic”, “total gastrectomy”, “surgical outcomes”, “complications”, “histopathology” and “health-related quality of life”. Reference lists from the identified studies were also screened to identify eligible additional studies not captured using the initial search. Editorials, letters, conference proceedings, case reports, preclinical and animal studies were excluded. The titles and abstracts of all short-listed articles were reviewed according to the established selection criteria and ineligible manuscripts excluded. The remaining publications were subjected to full-text review, and only those studies reporting postoperative outcomes of patients undergoing PTG were included. All included articles were available in English as full papers. A full search strategy has been included in the Appendix A. This systematic review adhered to the Preferred Reporting Items for Systematic review and Meta-Analysis (PRISMA) reporting guidelines and has not been registered with PROSPERO.

### 2.2. Data Extraction and Study Quality Assessment

The following clinical covariables were summarised across the studies: median age at surgery, patient sex, operative intent, operative approach, median operative time, total length of stay and complication and mortality rates. The major and minor postoperative complications were listed as reported in each study. For the majority of the studies, postoperative events were graded according to the Clavien–Dindo system, with grades 1 and 2 considered minor and grades 3 to 5 considered major. For the histopathological outcomes, the number of microscopic signet ring foci, median total number of lymph nodes examined and number of metastatic lymph nodes, tumour stage according to the eighth edition of the American Joint Committee on Cancer and the resection margin status were summarised. Lastly, for health-related QOL outcomes, persistent symptoms and psychosocial outcomes were included. To assess the risk of bias, observational studies were evaluated using the eight-point Newcastle–Ottawa scale (NOS) [11]. Briefly, one star for each satisfied item is assigned for the “Selection” and “Outcome” categories, while a maximum of two stars can be assigned for “Comparability”. Only studies that achieved either fair or good scores at 5 to 6 and 7 to 9 points, respectively, were included in this systematic review, with a higher number of stars indicating a higher study quality. Two authors were involved in the process, and a third author was available to resolve discrepancies that arose.

## 3. Results

### 3.1. Literature Search

Of the 257 articles screened from electronic searches and reference lists, 20 were included for the final review. Thirty articles were excluded due to search duplication. Based upon title and abstract reading, 177 articles were not relevant to the topic, leading to the full-text review of 30 articles, with 21 articles reporting outcomes and included in this review. A flowchart summarising the study selection is shown in Figure 1. All studies were retrospective cohort studies of levels 3 to 4 evidence [12]. The studies selected were of relatively high quality according to the NOS, where the majority achieved a score of 6 and above (Appendix A). A summary of the studies examining surgical, histopathological outcomes and health-related QOL outcomes is shown in Table 1, Table 2 and Table 3, respectively.

### 3.2. Surgical Outcomes of PTG

There were 15 studies reporting clinical outcomes following PTG in patients with *CDH1* pathogenic variants, with a total of 353 patients (Table 1) [13,14,15,16,17,18,19,20,21,22,23,24,25,26,27]. The median age at the time of surgery was 42 years old (32–54), and there was a greater proportion of female patients at 62.2%. The surgical approach described in the included studies was a total gastrectomy with Roux-en-Y oesophagojejunostomy reconstruction, with the majority via an open approach. There were four studies that reported the use of the laparoscopic approach, with no higher rate of postoperative complications reported compared to the open approach. Five studies reported the oesophagojejunal anastomotic type; three studies used an end-to-side circular stapler, while one study used a two-layer handsewn anastomosis, and another study used both end-to-side and side-to-side anastomosis in different patients. The median major and minor complication rates were 19.2% (IQR: 16.6–28.5%) and 10.5% (IQR: 10.0–15.0%), respectively, across 10 studies [14,15,17,19,20,21,22,23,24]. The most common complications reported pooled across these studies were wound infection at 4.8% and anastomotic leak and pulmonary complications at 4.5% each (Appendix A). The anastomotic stricture rate was 2.8% in seven patients. The 30-day postoperative mortality rate was 0.3% due to multiorgan failure secondary to sepsis in one patient with a history of kidney transplant and hepatitis. The type and rate of complications reported in the included studies are summarised in Appendix A.

### 3.3. Histopathological Outcomes of PTG

There were 19 studies that examined the histopathological outcomes following PTG in 414 patients (Table 2) [13,14,15,16,17,18,19,20,21,22,23,24,25,26,27,28,29,30,31]. Following PTG, 88.6% of patients had early lesions, including pagetoid 202 in situ lesions and foci of intramucosal signet ring cell adenocarcinoma. Among the patients with a positive final histopathology result, the mean/median number of SRC foci in each gastrectomy specimen ranged from less than 3 to 78. Out of 319 patients with positive pathological results, all cases were stage 1. There was no reported lymph node involvement in 12 studies, and all patients who underwent PTG achieved R0 resection. There were two studies that reported a need for reintervention due to incomplete removal of the gastric mucosa at the oesophageal margin in 3 out of 29 and 2 out of 6 patients [25,30].

### 3.4. Health-Related Quality of Life (QOL) following PTG

There were six studies that examined health-related QOL following PTG [20,21,26,31,32,33]. Questionnaires, including the European Organization for Research and Treatment for Cancer core QOL Questionnaire (EORTC QLQ C30), the gastric cancer specific module (EORTC QLQ STO22) and a 36-item short form health survey were utilised prior to and 1 year following surgery for one of the studies including 32 patients [31]. At baseline, there was no significant difference in mental health, depending on the *CDH1* mutation status and treatment preference. Physical functioning reduced after surgery but recovered to baseline by 12 months in three studies [21,31,32]. Similarly, mental functioning reduced in the first month after surgery but recovered by 3 to 9 months [31]. Similar findings were reported in three studies, where weight patterns stabilised 6 to 12 months postoperatively [21,31,32]. However, physical symptoms persisted after PTG, such as fatigue (75.5%) [31,32], dumping (61.5%) [21], pain (56.3%) [20,26,31], reflux (55.8%) [21,26,31] and diarrhoea (39.3%) [20,31,32]. Notably, small bowel bacterial overgrowth, pancreatic enzyme deficiency and bile acid malabsorption are common causes of these symptoms in this patient population [34]. The overall outcome was reported to be “as expected” by 40% of patients and “better than expected” by 45% [20]. A second study involved 27 patients who had undergone PTG and underwent qualitative interviews at a median of 3 years following surgery [33]. Most reported that undergoing surgery and convalescence was easier than anticipated. There was evidence that age affected the experiences of PTG, with younger patients tending to report faster recovery times and more transient aftereffects. All patients recognised the benefits of risk reduction as outweighing the costs of surgery. Nonetheless, surgery was reported to have a range of physical effects, including weight loss and fatigue, that impacted their psychological and social well-being.

## 4. Discussion

HDGC remains a rare condition accounting for less than 3% of patients with gastric cancer. There have been several small studies reporting on the surgical, pathological and health-related QOL outcomes. In this systematic review, we have synthesised the data across these studies to reveal a more detailed picture of these outcomes to aid informed consent.

Our review confirms that an extended lymphadenectomy is not necessary for an asymptomatic patient undergoing PTG, as lymph node metastasis was not identified in 360 patients across 18 studies. These findings support D1 lymphadenectomy as the standard approach, in line with the current guidelines, as the rate of lymph node metastasis is low and there is a higher reported complication rate associated with D2 lymphadenectomy [6]. Notably, the majority of the studies reported use of an open approach, and the outcomes from laparoscopic total gastrectomy might not have been adequately represented [35]. Nonetheless, recent trials have reported there was no difference in the postoperative and oncological efficacy outcomes between open and laparoscopic gastrectomy for gastric cancer [36]. Specifically, the outcomes demonstrated the noninferiority of laparoscopic surgery compared to open gastrectomy for both stage 1 and locally advanced gastric cancer. Beyond the potential applicability of a minimally invasive approach, other surgical technical elements, including positioning the Roux limb relative to the colonic mesentery, the use of a jejunal pouch, preservation of the posterior vagus nerve and the technique of the oesophagojejunal anastomosis, were not well described in the reported studies, and no further conclusion can be made about the impact of one technique over alternatives on clinical or quality of life outcomes.

For both prophylactic and screen-detected cases, it is essential to ensure the complete resection of the entire gastric mucosa. Prophylactic gastrectomy specimens may carry multiple foci of malignant cells diffusely distributed throughout the specimen where residual gastric tissue exposes the patient to the development of DGC [37]. Furthermore, determining the exact location of the squamocolumnar junction relative to the anatomical gastroesophageal junction is hindered by the lack of a palpable or visible surface markings. As such, the surgical removal of all gastric mucosa can be challenging. There were two studies that reported a need for reintervention due to incomplete removal of the gastric mucosa at the proximal resection margin [25,38]. In line with these findings, intraoperative confirmation of the squamous mucosa in the proximal oesophageal and duodenal mucosa in the distal margins is recommended in the international guidelines to ensure complete removal of the gastric mucosa [6]. Notably, not all of the studies in this systematic review reported if the entire gastric mucosa was assessed according to the standard Swiss roll technique, where the gastrectomy specimen should be fixed briefly for up to 3 h, and after which, the mucosa is dissected from the submucosa and muscle layers, as well as if all slides were examined, which may lead to an under-estimation of the SRC foci.

In view of the potential multifocal location of the SRC foci, this highlights the importance of performing a thorough and careful assessment of the gastric mucosa during endoscopic surveillance. This was similarly highlighted in this review, where the median percentage of a positive pathology result was 92.2% (IQR: 85.0–100.0%), despite no evidence of SRC foci on the endoscopic evaluation in 85.3% of patients. A recent prospective cohort study by Lee et al. assessed the detection rate of SRC in random biopsy samples taken according to the Cambridge protocol compared with biopsies targeted at endoscopic findings [37]. Notably, the first diagnosis of SRC was most commonly made from random biopsies in 50% of patients rather than targeted biopsies, and the anatomical distribution of SRC detected by random biopsies more accurately reflected those identified in PTG specimens. As such, random biopsies according to the Cambridge protocol improve the early detection of SRC, and this sampling method should be the standard of care for surveillance endoscopies [37]. On balance, approximately 1.5 targeted biopsy samples were taken for each endoscopy, and a recent commentary highlighted the low number of targeted biopsies after a 30-min inspection protocol in which samples should be taken from subtle mucosal changes [39]. In addition, the histological characteristics of the lesions were not reported in detail, and there might have been an underreporting of atypical lesions [39]. Hence, thorough inspection of the gastric mucosa with the standard use of narrow-band imaging and obtaining multiple targeted biopsy samples from mucosal abnormalities in conjunction with systematic random biopsies would contribute to an increased yield of SRC foci detection during endoscopic surveillance.

Our review highlights that, despite many patients proceeding to PTG at a young age, there is still an overall major postoperative complication rate of 19.2%. A total of 14 studies examined the perioperative surgical outcomes of total gastrectomy for prophylactic intent. The most common complications in this review included wound infection at 4.8% followed by anastomotic leak and pulmonary complications at 4.5% each. The largest single-centre experience reported for *CDH1* pathogenic carriers who underwent PTG was from the Memorial Sloan Kettering Cancer centre consisting of 101 patients [22]. The early complication rate was 16.0%, where wound infection and anastomotic stricture requiring dilatation were the most common at 8.0% and 5.0%, respectively. An additional study with 26 patients had a lower morbidity rate, where pulmonary complications and anastomotic leakage were 3.8% each [21]. The number of patients requiring a reintervention were similar in these studies, ranging from 9.0% to 27.0%. These outcomes are largely lower compared to studies examining complications associated with total gastrectomies in sporadic gastric cancer, where the 30-day morbidity and mortality rates were up to 29.3% and 5.4%, respectively [9]. This was likely due to the different demographics in both groups, where the patient population with sporadic gastric cancer were older, with a higher degree of comorbidity [40]. Nonetheless, in view of the potential significant risks of PTG, it is recommended this procedure should be performed at centres with extensive experience in gastric surgery. Hence, the IGCLC recommends that PTG be offered by centres performing at least 25 gastrectomies per year with a prospectively audited surgical 30-day mortality of less than 5% [7]. The prognostic outlook of this patient population appears to be promising, where none of the patients developed recurrence and with varying durations of follow-up ranging from 1 to 17 years [13,14,16,26]. Moving forward, the long-term survival and recurrence rates following PTG need to be formally evaluated in prospective studies.

Following a Roux-en-Y reconstruction food bypasses the duodenum and a variable length of the proximal jejunum, which are the major sites for calcium, iron and folate absorption [34]. Furthermore, a total gastrectomy results in the loss of intrinsic factor secretion and impaired vitamin B12 absorption [41]. Decreased intestinal transit time is also proposed to cause fat malabsorption in most patients, reducing the absorption of vitamins A, D, E and K [42]. Consequently, there are long-term metabolic and nutritional consequences of a total gastrectomy that have been reported in several studies, including calcium loss, which warrants regular bone density evaluation [20,21,26,30,31,43]. In addition, the most common chronic symptoms include appetite loss, fatigue and dumping, which are more severe in the early postsurgical period and decrease over time [28,31]. This is likely due to intestinal adaptation and altered eating habits in affected individuals. In studies reporting postoperative weight changes, most patients lost significant weight in the first 6 to 12 months, typically reaching, at the lowest, 15% to 23% of their initial weight [26]. Weight loss 1.5 to 2 years after surgery stabilised at 15% to 24% of the baseline body weight [21,26,30]. Notably, only two studies reported the use of a surgical jejunostomy; however, there was no evaluation on its impact on weight loss, which may be a clinically relevant area for future research [19,21]. These findings highlight the importance of involving regular follow-ups with a multidisciplinary team, including a dietician, and a comprehensive evaluation of the patient’s micronutrient levels to allow early intervention in the advent of deficiency.

One approach that has been proposed to reduce gastrointestinal symptoms is preservation of the coeliac branches of the posterior vagus at the time of PTG. A randomised controlled trial of vagus-preserving compared to conventional distal gastrectomy in 163 patients with sporadic gastric cancer demonstrated lower rates of diarrhoea and appetite loss at 12 months post-surgery; however, no long-term data exist to confirm the role of vagal preservation in patients undergoing PTG [44].

In addition to the physiological effects of PTG, studies have reported considerable psychosocial implications following surgery [30,31]. Although global physical and mental functioning recovered by 12 months post-surgery, there remained several aspects, including symptoms of dumping syndrome, eating restrictions and body image, that did not fully recover [20,21,26]. The full effect of these symptoms on overall QOL was not evident from the questionnaires used, and further examinations with qualitative methods may provide greater insight into patients’ experiences after gastrectomy. These studies have shown a wide range of psychosocial effects patients experience following PTG, which are closely related to their physical symptoms. The chronic nature of gastrointestinal symptoms, persistent weight loss and long-term nutritional deficiencies represent important issues to address in this population. These findings support the essential need for ample psychosocial support and holistic care. Moreover, in young patients, postponing surgery in favour of endoscopic surveillance could be better for their QOL, especially pertaining to mental issues, family and career planning. Conversely, many patients fear the development of DGC and experience significant anxiety around the time of surveillance endoscopy and awaiting their histological results. Moving forward, more studies evaluating the long-term health-related QOL in this patient population will help to promote informed and shared decision-making.

The main limitation of this systematic review is the absence of a registered protocol prior to data collection. Nonetheless, this systematic review adhered to the PRISMA guidelines (Preferred Reporting Items for Systematic Reviews and Meta-Analyses), which have been widely endorsed for the transparent and accurate reporting of systematic reviews and allow for replication of the study methods.

## 5. Conclusions

In conclusion, PTG remains the definitive treatment option for patients with pathogenic *CDH1* germline variants who are appropriate surgical candidates. It is imperative for patients to receive comprehensive preoperative counselling to make an informed decision through a shared decision-making process with their clinician. There are significant clinical and psychosocial implications of undergoing PTG that highlight the need for holistic patient management. Alternatively, there is validity in endoscopic surveillance for selected patients wishing to defer surgery in centres with appropriate endoscopic expertise. This systematic review also demonstrated there is potential significant morbidity in PTG, but it is safe and oncologically sound. The areas ripe for research are to define when the optimal time is for patients to have surgery if endoscopic surveillance is available; which surgical approach should be used to reduce postoperative complications and if surgical techniques are able to reduce some of the later effects, such as a minimally invasive approach, vagal-sparing gastrectomy and the use of a jejunal pouch. Furthermore, the long-term survival, reintervention rates and QOL remain to be established. Moving forward, further evaluation is needed to examine the long-term outcomes and survival outlook of patients who undergo PTG.

## Figures and Tables

**Figure 1 cancers-16-00473-f001:**
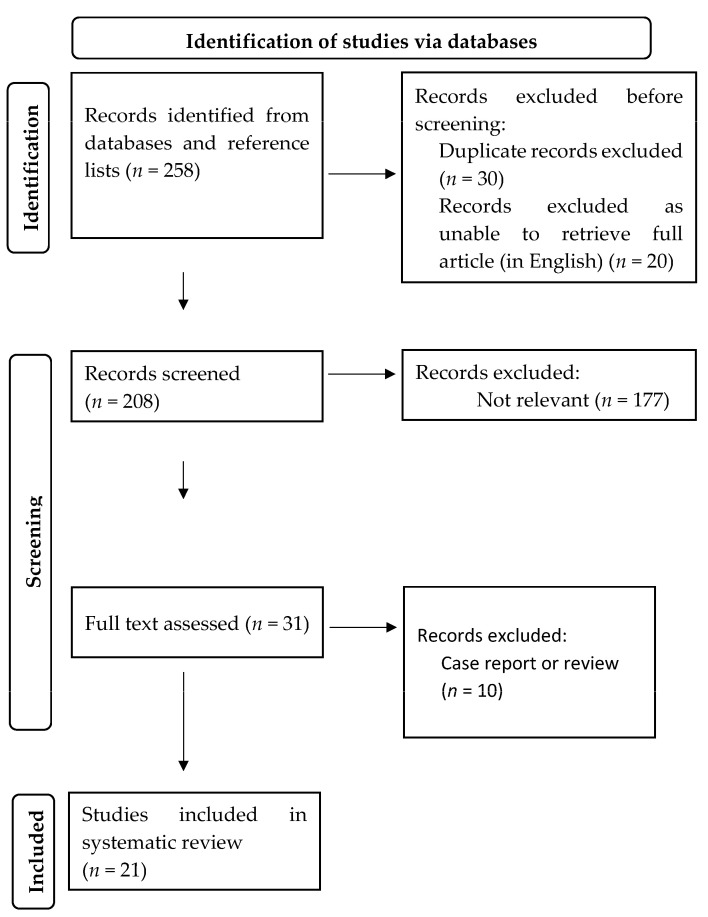
Flowchart for the selection of the studies included in the systematic review.

**Table 1 cancers-16-00473-t001:** Surgical outcomes following total gastrectomy in patients with pathogenic *CDH1* variants.

No.	First Author	Study Year	No. of Patients *n*	Mean/ Median Age (Range)	Female Patients *n* (%)	Male Patients *n* (%)	Endoscopy Protocol	Median Length of Stay (Days) (Range)	Minimally Invasion Surgery (*n*)	Open Approach *n* (%)	Oesophagojejunostomy Anastomotic Type	Median Operative Time (Minutes) (Range)	Postoperative Complication Rate *n* (%)		Complication Requiring Reintervention *n* (%)	Mortality Rate (%)	Reference
													Major	Minor			
1	Norton JA	2007	6	54 (51–57)	4 (66.7%)	2 (33.3%)	Chromoendoscopy	7 (6–10)	0	6 (100.0%)	-	-	0	0	0	0	[13]
2.	Hebbard PC	2009	23	45 (26–63)	14 (61.0%)	9 (39.0%)	-	11 (9–107)	0	23 (100.0%)	-	-	4 (17.4%)	7 (30.4%)	2 (9.0%)	0	[14]
3.	Pandalai PK	2011	10	44 (27–52)	4 (40.0%)	6 (60.0%)	-	7 (7–8)	0	10 (100.0%)	Handsewn in 2 layers	234 (187–308)	3 (30.0%)	1 (10.0%)	0	0	[15]
4.	Chen Y	2011	13	45 (18–70)	9 (69.2%)	4 (30.8%)	-	6	0	13 (100.0%)	-	-	0	0	0	0	[16]
5.	Hackenson D	2011	6	42 (27–51)	4 (40%)	6 (60%)	-	7 (7–8)	0	6 (100.0%)	Circular stapler (OrVil)	213 (187–308)	2 (33.3%)	0	0	0	[17]
6.	Bardram L	2014	7	41 (26–52)	3 (42.9%)	4 (57.1%)	Cambridge	7 (6–8)	0	7 (100.0%)	-	-	0	0	0	0	[18]
7.	Haverkamp L	2015	11	40 (22–61)	8 (72.7%)	3 (27.3%)	-	10 (7–27)	10 (90.9%)	1 (9.1%)	Circular stapler (OrVil), distal jejunum used to create “J”-shaped pouch	266 (217–315)	2 (18.2%)	2 (18.2%)	1 (9.1%)	0	[19]
8.	Strong VE	2016	41	47 (20 to 71)	27 (65.9%)	14 (34.1%)	-	7 (4 to 50)	16 (39%)	25 (61%)	-	197 (115–356)	11 (27.0%)	0	11 (27.0%)	2.5%	[20]
9.	Van der Kaaij RT	2018	26	41 (30 to 53)	14 (53.8%)	12 (46.2%)	Cambridge	-	0	26 (100.0%)	-	167 (147–194)	5 (19.2%)	3 (11.5%)	5 (19.2%)	0	[21]
10.	Vos EL	2020	101	42	68 (67.3%)	31 (30.7%)	White light endoscopy and NBI	7 (6–7)	43 (42.6%)	58 (57.4%)	-	193 (163–218)	16 (15.8%)	12 (11.8%)	16 (16.0%)	1.0%	[22]
11.	Devezas V	2020	19	39 (14–63)	9 (47.4%)	10 (52.6%)	Cambridge	6	4 (21.1%)	15 (78.9%)	-	-	0	2 (10.5%)	0	0	[23]
12.	DiBrito SR	2020	8	48 (24–51)	6 (75.0%)	2 (25.0%)	-	7 (6–20)	0	8 (100.0%)	Jejunal pouch creation using linear stapler	246 (161–414)	2 (20.0%)	1 (10.0%)	2 (20.0%)	0	[24]
13.	Lewis FR	2021	6	32 (22–40)	4 (66.7%)	2 (33.3%)	-	7	0	6 (100.0%)	2 end-to-side handsewn and 2 stapled, 2 side-to-side with linear stapler	-	-		0	0	[25]
14.	Forrester JD	2022	22	37 (27–71)	12 (55.1%)	11 (44.9%)	Chromoendoscopy	5	0	22 (100.0%)	-	-	0	0	0	0	[26]
15.	Stillman MD	2022	54	41 (16–70)	35 (64.8%)	19 (35.2%)	-	7	0	54 (100.0%)	10 end-to-side handsewn, 44 side-to-side with linear stapler	161 (116–308)	2 (3.7%)	3 (5.6%)	0	0	[27]

NBI: Narrow band imaging.

**Table 2 cancers-16-00473-t002:** Histopathological outcomes of total gastrectomy specimens.

No.	First Author	Study Year	Total No. of Patients Who Underwent PTG (*n*)	No. of Patients with Pathological Malignancy (%)	Median (*)/Mean (#) No. of Microscopic Foci of Signet Ring Carcinoma (Range)	Stage				No. of Patients with Positive Lymph Nodes (%)	Median No. of Lymph Nodes Harvested (*n*)	Median No. of Positive Lymph Nodes (range)	No. of Patients with Proximal Squamous Margin (%)	No. of R0 Resection (%)	Reference
						1	2	3	4						
1.	Lewis FR	2001	6	6 (100.0%)	-	6 (100.0%)	0	0	0	0	-	-	4 (66.6%)	6 (100.0%)	[25]
2.	Charlton A	2004	6	6 (100.0%)	78 (4–318) *	6 (100.0%)	0	0	0	0	15 (6–26)	0	-	6 (100.0%)	[28]
3.	Norton JA	2007	6	6 (100.0%)	-	6 (100.0%)	0	0	0	0	24 (18–40)	0	-	6 (100.0%)	[13]
4.	Rogers WM	2008	8	8 (100.0%)	10.9 (2–17) *	8 (100.0%)	0	0	0	0	18 (7–41)	0	-	8 (100.0%)	[29]
5.	Hebbard PC	2009	23	22 (96.0%)	N.A. (1–52)	22 (96.0%)	0	0	0	0	-	-	23 (100.0%)	22 (100.0%)	[14]
6.	Hackenson D	2010	6	6 (100.0%)	-	6 (100.0%)	-	0	0	0	10.5 (10–25)	0	-	6 (100.0%)	[17]
7.	Pandalai PK	2011	10	9 (90.0%)	-	9 (90.0%)	0	0	0	0	12	0	-	9 (100.0%)	[15]
8.	Chen Y	2011	13	17 (94.4%)	-	12 (82.3%)	-	0	0	0	-	-	-	13 (100.0%)	[16]
9.	Klujit I	2012	29	27 (93.1%)	-	29 (100.0%)	-	-	-	-	-	-	26 (89.7%)	29 (100.0%)	[30]
10.	Bardram L	2014	7	7 (100.0%)	25.5 (5–82) *	7 (100.0%)	0	0	0	0	16 (12–41)	0	7 (100.0%)	7 (100.0%)	[18]
11.	Haverkamp L	2015	11	9 (81.8%)	-	9 (100.0%)	-	0	0	0	10 (1–25)	0	-	9 (100.0%)	[19]
12.	Worster E	2016	18	9 (69.2%)	-	9 (69.2%)	0	0		0	25 (9–58)	0	-	9 (100.0%)	[31]
13.	Strong VE	2017	41	35 (85%)	-	35 (100.0%)	0	0	0	0	-	-	-	35 (100.0%)	[20]
14.	Van der Kaaij RT	2018	26	23 (88.5%)	6 (2–12) *	23 (88.5%)	-	-	-	-	19 (13–22)	0	26 (100.0%)	23 (100.0%)	[21]
15.	Vos EL	2020	101	88 (87.1%)	-	88 (87.1%)	-	-	-	0	13 (18–26)	0	-	88 (100.0%)	[22]
16.	Devezas V	2020	19	17 (89.5%)	6 (1–37) *	12 (63.2%)	0	0	0	0	22 (7–36)	0	-	12 (70.6%)	[23]
17.	DiBrito SR	2020	8	3 (37.5%)	-	3 (37.5%)	0	0	0	0	13 (6–18)	0	8 (100.0%)	8 (100.0%)	[24]
18.	Forrester JD	2022	22	21 (95.5%)	13 (2–23) #	21 (95.5%)	0	0	0	0	-	-	-	22 (100.0%)	[26]
19.	Stillman MD	2022	54	52 (96.3%)	15 (5–136) * for 10 patients 2 (0–5) * for 44 patients	52 (96.3%)	0	0	0	0	-	0	54 (100.0%)	54 (100.0%)	[27]

Median no. of microscopic foci of signet ring carcinoma is indicated by *. Mean no. of microscopic foci of signet ring carcinoma is indicated by #. PTG: prophylactic total gastrectomy.

**Table 3 cancers-16-00473-t003:** Health-related quality of life outcomes following total gastrectomy.

No.	First Author	Study Year	No. of Patients (*n*)	No. of Female Patients (*n*)	No. of Male Patients (*n*)	Median Age (Years)	Median Weight Loss (%/kg)		Symptoms									Reference ^
							6 Months	1 Year	Appetite Loss *n* (%)	Dyspnoea *n* (%)	Fatigue *n* (%)	Pain *n* (%)	Diarrhoea *n* (%)	Dumping *n* (%)	Eating Restriction *n* (%)	Reflux *n* (%)	Body Image *n* (%)	
1.	Worster E	2014	32	17	15	35 (16–64)	-	18.0%	-	-	63.0%	26 (81.0%)	22 (70.0%)	-	45.0%	20 (63.0%)	14 (44.0%)	[31]
2.	Muir J	2016	13	11	2	51	17 kg	-	8 (63.0%)	8 (63.0%)	11 (88.0%)		5 (38.0%)	-	-	-	-	[32]
3.	Strong VE	2016	41	27	14	47 (20 to 71)	-	-	-	-	-	18 (45.0%)	4 (10.0%)	-	-	-	-	[20]
4.	Van der Kaaij RT	2018	26	14	12	-	12.0%	15.0%	-	-	-	-	-	16 (61.5%)	-	16 (61.5%)	-	[21]
5.	Forrester JD	2022	30	17	39 (27–71)		-	23.0%	-	-		13 (43.0%)	-	-	--	13 (43.0%)	-	[26]
									Symptom rate across studies (%)									
									-	-	68.9%	55.3%	36.0%	-	-	55.7%	-	

^ Study reference [32] was a qualitative study; hence, it is not included in Table 3.

## Data Availability

No new data were created or analyzed in this study. Data sharing is not applicable to this article.

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
