# Peer review of "A Systematic Review on Clinical and Health-Related Quality of Life Outcomes following Total Gastrectomy in Patients with Hereditary Diffuse Gastric Cancer"

_cancers, 2024, doi:10.3390/cancers16030473_

Round 1

Reviewer 1 Report

Comments and Suggestions for Authors

The manuscript by Lim et al., titled "A Systematic Review on Clinical and Health-related Quality of Life Outcomes following Total Gastrectomy in Patients with Hereditary Diffuse Gastric Cancer," presents a comprehensive study on a vital topic. To enhance the clarity and impact of your research, consider the following suggestions:

1. Clarify the Study's Objective and Importance:

   Clearly articulate the primary objective of your systematic review. Explain why evaluating clinical and health-related quality of life outcomes in Hereditary Diffuse Gastric Cancer (HDGC) patients undergoing prophylactic total gastrectomy (PTG) is crucial. Providing this context will underscore the significance of your research.

2. Include Key Findings:

   Highlight key findings or emerging trends from your systematic review. For instance, mention prevalent complications, median patient age, and the proportion of patients with early lesions post-PTG. These details offer readers a preview of your results, piquing their interest to explore the complete paper.

3. Emphasize Psychosocial Impact:

   Strengthen the focus on the psychosocial impact of PTG. Elaborate on the diverse range of psychosocial effects, linking them to specific physical symptoms or complications. This detailed exploration will underscore the necessity of comprehensive pre-operative counseling.

4. Highlight Implications and Recommendations:

   Devote a section to discussing the implications of your findings. Explain the significance of these outcomes for clinical practices and patient care. Additionally, provide recommendations for healthcare providers and clinicians dealing with HDGC patients undergoing PTG. Concrete suggestions enhance the practical utility of your research.

5. Cite Relevant Publications:

   Support your study by citing pertinent publications, studies, or statistics related to HDGC and PTG. These citations will serve as quick reference points, contextualizing your research within the existing body of knowledge.

6. Review Grammar and Spelling:

   Thoroughly proofread the manuscript to rectify any grammatical or spelling errors. Ensure that the language used is precise, clear, and adheres to the highest standards of academic writing.

7. Table Formatting:

   Revise the formatting of all tables for clarity. Ensure that the table numbers and descriptions are distinctly visible within the tables. Clear labeling will enhance understanding for readers.

By addressing these suggestions, the manuscript can significantly improve its overall quality. This will create a compelling and informative introduction, encouraging readers to thoroughly engage with the complete manuscript and gain a comprehensive understanding of your research.

Comments on the Quality of English Language

NA

Author Response

Thank you very much for the comments. Please find below response to the comments.

  1. Clarify the Study's Objective and Importance:

Clearly articulate the primary objective of your systematic review. Explain why evaluating clinical and health-related quality of life outcomes in Hereditary Diffuse Gastric Cancer (HDGC) patients undergoing prophylactic total gastrectomy (PTG) is crucial. Providing this context will underscore the significance of your research.

The aim of the systematic review has been clearly stated at the end of the introduction after providing background of the current literature guidelines and practice for HDGC patients (page 2, line 69 to 72).

  1. Include Key Findings:

Highlight key findings or emerging trends from your systematic review. For instance, mention prevalent complications, median patient age, and the proportion of patients with early lesions post-PTG. These details offer readers a preview of your results, piquing their interest to explore the complete paper.

The key findings have been detailed in the results and discussion section, including patient demographics, complications and pathological outcomes.

  1. Emphasize Psychosocial Impact:

Strengthen the focus on the psychosocial impact of PTG. Elaborate on the diverse range of psychosocial effects, linking them to specific physical symptoms or complications. This detailed exploration will underscore the necessity of comprehensive pre-operative counselling.

The psychosocial impact of PTG on various aspects of quality of life from various studies has been discussed on page 13.

  1. Highlight Implications and Recommendations:

Devote a section to discussing the implications of your findings. Explain the significance of these outcomes for clinical practices and patient care. Additionally, provide recommendations for healthcare providers and clinicians dealing with HDGC patients undergoing PTG. Concrete suggestions enhance the practical utility of your research.

The significance of our findings and recommendations on clinical practice have been detailed in the conclusion section (page 13).

  1. Cite Relevant Publications:

 Support your study by citing pertinent publications, studies, or statistics related to HDGC and PTG. These citations will serve as quick reference points, contextualizing your research within the existing body of knowledge.

Relevant studies evaluating outcomes following PTG in HDGC patients have been included in the systematic review and cited as references. The search strategy has been included in supplementary materials (S1) (page 16 to 17).

  1. Review Grammar and Spelling:

Thoroughly proofread the manuscript to rectify any grammatical or spelling errors. Ensure that the language used is precise, clear, and adheres to the highest standards of academic writing.

The manuscript has been proofread for grammar and spelling while ensuring language is clear and precise.

  1. Table Formatting:

Revise the formatting of all tables for clarity. Ensure that the table numbers and descriptions are distinctly visible within the tables. Clear labelling will enhance understanding for readers.

Table numbers and descriptions are clearly visible within tables.

Reviewer 2 Report

Comments and Suggestions for Authors

Authors investigated the clinical, histopathological or health-related quality of life outcomes in HDGC patients undergoing PTG, and they concluded that PTG remains the definitive treatment option for patients with pathogenic CDH1 germline variants who are appropriate surgical candidates(page.11, line.291-292). 

The materials were high volume and many factors were analyzed , however there was a queston  as below to reach the conclusion.

Authors didn't analyzed another surgical method such as partial gastrectomy. Was the prognosis and QOL of PTG better than those of partial gastrectomy?

Authors were recommended to compare prognosis and QOL of PTG with those of partial gastrectomy respectively to conclude that 'PTG remains the definitive treatment option'.  

Author Response

Thank you very much for the comments. Please find below response to comments.

Authors investigated the clinical, histopathological or health-related quality of life outcomes in HDGC patients undergoing PTG, and they concluded that PTG remains the definitive treatment option for patients with pathogenic CDH1 germline variants who are appropriate surgical candidates (page.11, line.291-292). 

The materials were high volume and many factors were analyzed, however there was a question as below to reach the conclusion.

  1. Authors didn't analyzed another surgical method such as partial gastrectomy. Was the prognosis and QOL of PTG better than those of partial gastrectomy? Authors were recommended to compare prognosis and QOL of PTG with those of partial gastrectomy respectively to conclude that 'PTG remains the definitive treatment option'.  

The high lifetime risk of diffuse-type gastric cancer in patients with HDGC appears to be eliminated when all gastric epithelium is removed.  For this reason, the recommended standard of care for prophylactic gastrectomy for patients with HDGC patients is to undergo a total gastrectomy.7 Hence, no clinical studies have been included reporting sub-total gastrectomy.

Reviewer 3 Report

Comments and Suggestions for Authors

Thank you for inviting me to peer review this systematic review focused on prophylactic total gastrectomy in patients with hereditary diffuse gastric cancer. While the topic is of importance I have concerns about the rigour of the review undertaken. My main concern relates to the fact there appears to be no reference to an a-priori protocol having been conducted. Without a protocol there are massive implications for the rigour, transparency and integrity of the review. I would recommend the authors provide evidence of a publicly available protocol, otherwise this review should be rejected. Without a protocol, it is unclear whether the authors deviated from what they had planned to focus on.

My individual comments are provided below and as you can see they focus on methods due to the absence of a protocol as well as on how the results are presented:

·       Exclusion criteria are not clear to me, they should be reported in full. ·       Why wasn't PROSPERO searched? ·       It is not clear which methodology was followed for the review, e.g., Cochrane, JBI? ·       Were any language restriction applied? I can see that language restriction was applied from Figure 1. The authors should clearly state what they did and discuss potential implications. ·       Authors should provide full search strategies as appendix for transparency and reproducibility purposes.

·       How many researchers participated in screening, data extraction, and quality assessment? Good practice requires at least two independent researchers engaged throughout the entire process and a third one should be available to solve discrepancies.

·       In figure 1, what does records excluded because not relevant mean? I would have expected to see the list of exclusion criteria and the number of studies excluded for each criterion. A protocol would outline the specific inclusion and exclusion criteria.

·       In table 1 first column, it would be useful to have the first author of the included studies rather than a progressive order with reference to the study in the last column.

·       How were major and minor post-operative complications defined?

·       At lines 96, the authors stated that all included studies were retrospective cohort studies, then the note under table 3 reports “*Study reference 32 was a qualitative study, hence it was not included in Table 3.” Which type of studies were included?

Author Response

Thank you very much for the comments. Please find below response to comments.

Thank you for inviting me to peer review this systematic review focused on prophylactic total gastrectomy in patients with hereditary diffuse gastric cancer. While the topic is of importance I have concerns about the rigor of the review undertaken. My main concern relates to the fact there appears to be no reference to an a-priori protocol having been conducted. Without a protocol there are massive implications for the rigor, transparency and integrity of the review. I would recommend the authors provide evidence of a publicly available protocol, otherwise this review should be rejected. Without a protocol, it is unclear whether the authors deviated from what they had planned to focus on.

My individual comments are provided below and as you can see, they focus on methods due to the absence of a protocol as well as on how the results are presented:

  1. Exclusion criteria are not clear to me, they should be reported in full. Why wasn't PROSPERO searched? It is not clear which methodology was followed for the review, e.g., Cochrane, JBI?       

The exclusion criteria have been included in the methods section (page 2, line 85 to 87) and search strategy explicitly detailed in supplementary materials to allow replication of our method. The systematic review adhered to the PRISMA guidelines (Preferred Reporting Items for Systematic Reviews and Meta-Analyses).  We thank the reviewer for highlighting the PROSPERO prospective register of systematic reviews which is a valuable resource.  We feel the existing literature would be successfully captured by our search strategy including existing systematic reviews on the topic.  Our understanding is that PROSPERO would not include a literature review using a systematic search employed in this paper.      

  1. Were any language restriction applied? I can see that language restriction was applied from Figure 1.

Only full text articles in English were included where a minority of articles were in other languages.

  1. Authors should provide full search strategies as appendix for transparency and reproducibility purposes.

A full search strategy has been included in Supplementary material S1.

  1. How many researchers participated in screening, data extraction, and quality assessment? Good practice requires at least two independent researchers engaged throughout the entire process and a third one should be available to solve discrepancies.

2 authors were involved in the process and a third author was available to resolve discrepancies which arose.

  1. In figure 1, what does records excluded because not relevant mean? I would have expected to see the list of exclusion criteria and the number of studies excluded for each criterion. A protocol would outline the specific inclusion and exclusion criteria.

Inclusion and exclusion criteria have been included in the methods section (page 2, line 75 to 87).

  1. In table 1 first column, it would be useful to have the first author of the included studies rather than a progressive order with reference to the study in the last column.

The first author for included studies have been listed in table 1 to 3.

  1. How were major and minor post-operative complications defined?

The major and minor post-operative complications were listed as reported in each study. For majority of studies, post-operative events were graded according to the Clavien-Dindo system with grades 1 and 2 considered minor and grades 3 to 5 considered major.

  1. At lines 96, the authors stated that all included studies were retrospective cohort studies, then the note under table 3 reports “*Study reference 32 was a qualitative study, hence it was not included in Table 3.” Which type of studies were included?

All studies were retrospective studies which included both quantitative and qualitative studies.

Reviewer 4 Report

Comments and Suggestions for Authors

The authors have reviewed postoperative and QOL outcomes examining existing papers on prophylactic gastrectomy for HDGC. Overall this is an interesting paper and adds to the literature. The methods are well describes and the paper is well written. I have some minor comments:

1) Summary line 24. Stage 1 on stage 0? please define in your methods what staging system you are using. Please correct on the text as appropriate 

2) Methods: I am not familiar with the Newcastle-Ottawa scale so cannot comment on the use of it 

3) Table 1. You are can erase either the female or the male column as It is not necessary to give both 

4) Table 1 needs some careful minor editing for example column MIS approach: MIS needs to be spelled out and since you are giving both n and percentage you should type is as  "n(%) " 

5) Table 1. Any data on stricture formation long term in the included studies?

6) Section 3.2 Please give the causes of postoperative mortality. This is especially important for a procedure that has prophylactic purpose 

7) Discussion. Please provide reference in line 176. Also consider mentioning conclusions from the KLASS trials on laparoscopic vs open especially postoperative outcomes 

8) Discussion line 220. Please elaborate on. mortality 

9) Discussion line 246. Any data from the included studies on the length of Roux limb and if it is related to nutritional deficiencies?

10) Conclusions. This paragraph needs revision and shortening as conclusions should be drawn only from the analyzed data. For example the sentence that there is validity of endoscopic surveillance is not supported by the data of this study. You can add an appropriate reference in the discussion part to comment on alternative strategies

Comments on the Quality of English Language

See above

Author Response

Thank you very much for the comments. Please find below response to comments.

The authors have reviewed postoperative and QOL outcomes examining existing papers on prophylactic gastrectomy for HDGC. Overall this is an interesting paper and adds to the literature. The methods are well describes and the paper is well written. I have some minor comments:

  1. Summary line 24. Stage 1 on stage 0? please define in your methods what staging system you are using. Please correct on the text as appropriate.

The staging system has been included in methods section 2.2 (page 2 and 3, line 93 to 94) and all cases with evidence of at least intramucosal cancer in the resection specimen were stage 1 (pT1aN0).

  1. Table 1. You can erase either the female or the male column as it is not necessary to give both.

Both the number of female and male patients were included in table 1 to ensure complete information was provided.

  1. Table 1 needs some careful minor editing for example column MIS approach: MIS needs to be spelled out and since you are giving both n and percentage you should type is as "n (%) ".

MIS has been spelled out. Both n and percentage have been included in table 1.

  1. Table 1. Any data on stricture formation long term in the included studies?

The incidence of anastomotic stricture has been provided in 3 studies (reference 15, 17 and 22) (page 10, line 156) with further details in supplementary table S3.

  1. Section 3.2 Please give the causes of postoperative mortality. This is especially important for a procedure that has prophylactic purpose.

The causes of the post-operative mortality have been included in results section 3.2 (page 10, line 154 to 155). 

  1. Please provide reference in line 176. Also consider mentioning conclusions from the KLASS trials on laparoscopic vs open especially postoperative outcomes.

A reference has been included in original line 176 (page 11, line 208 and 209). KLASS trial was a comparison between laparoscopic and open approach for distal gastrectomy and is not relevant to our review.  The LOGICA and CLASS-02 trials are more relevant, and they have reported no significant difference in complications or oncological outcomes between both approaches (page 11, line 207 to 210).

  1. Discussion line 220. Please elaborate on mortality.

The causes of the post-operative mortality have been included in results section 3.2 (page 10, line 154 to 155).

  1. Discussion line 246. Any data from the included studies on the length of Roux limb and if it is related to nutritional deficiencies?

There is no specific data or systematic approach in included studies on association of the length of Roux limb with nutritional deficiencies. Nonetheless, literature from bariatric studies have shown importance of roux limb length in weight loss. Despite likely efforts to minimise the roux limb length in the included studies, the median weight loss at 1 year ranges from 15% to 23%, reflecting this an important aspect of quality of life.

Reviewer 5 Report

Comments and Suggestions for Authors

The review article “A Systematic Review on Clinical and Health-related Quality of Life Outcomes following Total Gastrectomy in Patients with Hereditary Diffuse Gastric Cancer” is an interesting article that mainly is the compilation of the clinical and health-related quality of life outcomes following the prophylactic total gastrectomy (PTG) for Hereditary diffuse gastric cancer (HDGC) patients.

Authors have discussed the HDGC syndrome, an autosomal dominant condition predominantly attributed to germline loss-of-function mutations in the tumor suppressor gene CDH1 involved in epithelial cell adhesion.

They considered the studies published from Jan 2000 to Dec 2022, selected 21 articles (out of 257), and included only those studies that reported the post-operative outcomes of PTG patients among other selection criteria. A good flowchart is made for the selection of studies as shown in Figure 1. Surgical outcomes following gastrectomy, histopathological outcomes, and health-related quality of life outcomes are presented in tabular form along with details in the text form.

Overall, the article is well written both in terms of the context and the English language. The data from several studies on surgical, pathological, and health-related QOL has been presented in a comprehensive way. The authors have discussed the article in a well-structured manner with proper references. It covers the small gaps related to the topic and concludes that PTG remains the definitive treatment option for patients with pathogenic CHD1 mutation. The authors also emphasized following the informed decision-making process and other alternative options for such patients to reduce post-operative complications.

A long-term and multi-institutional prospective study is needed to come up with better statistical results related to the topic which can help in better decision-making at different stages of disease development.

There are some minor mistakes in the sentences like in line 173. Line 69, the sentence is incomplete. The authors need to carefully read the manuscript to correct such mistakes.

Authors should decrease the font size or use other alternatives to make the tables more readable. The contents of the table look scattered and difficult to read.

I would recommend the acceptance of the article after the suggested minor corrections are made.

Comments on the Quality of English Language

The English language is OK. Just some minor errors are present that need correction.

Author Response

Thank you very much for the comments. Please find below response to comments.

The review article “A Systematic Review on Clinical and Health-related Quality of Life Outcomes following Total Gastrectomy in Patients with Hereditary Diffuse Gastric Cancer” is an interesting article that mainly is the compilation of the clinical and health-related quality of life outcomes following the prophylactic total gastrectomy (PTG) for Hereditary diffuse gastric cancer (HDGC) patients.

Authors have discussed the HDGC syndrome, an autosomal dominant condition predominantly attributed to germline loss-of-function mutations in the tumor suppressor gene CDH1 involved in epithelial cell adhesion.

They considered the studies published from Jan 2000 to Dec 2022, selected 21 articles (out of 257), and included only those studies that reported the post-operative outcomes of PTG patients among other selection criteria. A good flowchart is made for the selection of studies as shown in Figure 1. Surgical outcomes following gastrectomy, histopathological outcomes, and health-related quality of life outcomes are presented in tabular form along with details in the text form.

Overall, the article is well written both in terms of the context and the English language. The data from several studies on surgical, pathological, and health-related QOL has been presented in a comprehensive way. The authors have discussed the article in a well-structured manner with proper references. It covers the small gaps related to the topic and concludes that PTG remains the definitive treatment option for patients with pathogenic CHD1 mutation. The authors also emphasized following the informed decision-making process and other alternative options for such patients to reduce post-operative complications.

A long-term and multi-institutional prospective study is needed to come up with better statistical results related to the topic which can help in better decision-making at different stages of disease development.

  1. There are some minor mistakes in the sentences like in line 173. Line 69, the sentence is incomplete. The authors need to carefully read the manuscript to correct such mistakes.

The sentence has been corrected, and manuscript proofread for grammar and spelling errors.

  1. Authors should decrease the font size or use other alternatives to make the tables more readable. The contents of the table look scattered and difficult to read.

The font size has been decreased and the tables have been placed in landscape format to make them easier to read.

Round 2

Reviewer 1 Report

Comments and Suggestions for Authors

present form is acceptable

Comments on the Quality of English Language

present form is acceptable

Author Response

Thank you very much for the feedback.

Reviewer 2 Report

Comments and Suggestions for Authors

Authors didn't respond reviewers recommendation.  Authors only selected PTG as the subjects , therefore the method couldn't be estimated as the best treatment by the subjects. 

Author Response

Thank you very much for the comments. The systematic review only included patients who had no pre-operative evidence of established diffuse gastric cancer and underwent prophylactic total gastrectomy (PTG) which is the case for majority of HDGC patients. Furthermore, the high lifetime risk of diffuse-type gastric cancer in patients with HDGC appears to be eliminated when all gastric epithelium is removed. For this reason, the recommended standard of care from the International Gastric Cancer Linkage Consortium (IGCLC) guidelines for patients with HDGC undergoing prophylactic surgery is to undergo a total rather than subtotal gastrectomy. Hence, no clinical studies have been included reporting sub-total gastrectomy.

Reviewer 3 Report

Comments and Suggestions for Authors

The authors have only partially addressed my previous comments. Beyond not clarifying the screening, data extraction, and quality assessment processes, not declaring language limitations in the main text, and not providing detailed information about the number of studies excluded for each criterion in figure 1, my main concern is the absence of an apriori protocol. Developing a protocol prior to conducting a systematic review is an internationally agreed upon step in systematic review methodology. This essentially ensures transparency in terms of certifying the authors did what they said they would in terms of their review question, inclusion criteria and methods. Without a protocol we can’t be sure that the authors did not deviate which makes the review open to bias, reducing its trustworthiness. Organisations such as Cochrane, JBI and Campbell require an apriori protocol – they will not consider a review without it. If this is not a requirement of the Journal I would suggest that this be reconsidered. The comment above still stands and I would still not feel comfortable in recommending the manuscript if a protocol was not developed.

Comments on the Quality of English Language

Minor editing of English language required

Author Response

Thank you very much for the comments. The screening (page 2, line 75 to 84), data extraction (page 2 and 3, line 92 to 101), and quality assessment processes have been detailed in the methods section. Furthermore, exclusion criteria have been included (page 3, line 102 to 108) and a full search strategy explicitly detailed in supplementary materials to allow replication of our method. Language limitations have been stated in the main text (page 2, line 84 to 85) and the number of studies excluded for each criterion in Figure 1 as suggested. Moreover, the systematic review adhered to the PRISMA guidelines (Preferred Reporting Items for Systematic Reviews and Meta-Analyses) which has been widely endorsed for transparent and accurate reporting of systematic reviews and allows for replication of study methods. The reviewer makes a valid point around registering systematic review protocols prior to data extraction.  We agree this should become more standardised in the literature and would do so for future work, although we have not on this occasion.  By providing detailed reporting of our search strategy that allows replication, we feel there is not a significant risk of bias in our study.  We also feel our search was broad enough and abstract review rigorous enough that no papers reporting significant post-operative outcomes of patients undergoing prophylactic gastrectomy have been missed. If any have been, we would be delighted to include them in this work and welcome the reviewer to provide us with the study references.

Round 3

Reviewer 2 Report

Comments and Suggestions for Authors

Authors answered again that ' For this reason, the recommended standard of care from the International Gastric Cancer Linkage Consortium (IGCLC) guidelines for patients with HDGC undergoing prophylactic surgery is to undergo a total rather than subtotal gastrectomy.'  Then , the conclusion of this article of ' PTG as best treatment' is seemed to be already well known and not novel. Therefore originality as article was low. 

Author Response

Thank you very much for the comments. The objective of our systematic review was, as stated in the aim, to summarise the outcomes of prophylactic total gastrectomy (PTG) in hereditary diffuse gastric cancer patients. As no other work has been done so far in this area, we feel this systematic review is novel and adds to the literature in this field to demonstrate there is potential significant morbidity in PTG, but it is safe and oncologically sound.

Reviewer 3 Report

Comments and Suggestions for Authors

The absence of a registered systematic review protocol should be at least declared as a main limitation of this study and the potential impact on study findings discussed. However, I continue to not feel comfortable in recommending the publication of manuscripts if a protocol was not developed. 

Comments on the Quality of English Language

Minor editing of English language required

Author Response

Thank you very much for the comments. The limitation of a registered systematic review protocol has been included in the discussion section (page 13, line 328 to 332). PROSPERO only accepts reviews provided data extraction has not yet started. By providing detailed reporting of our search strategy that allows replication, we feel there is not a significant risk of bias in our study. We feel our search was broad enough and abstract review rigorous enough that no papers reporting significant post-operative outcomes of patients undergoing prophylactic gastrectomy have been missed.